# Antibiotic Resistance, Biofilm Formation, and Presence of Genes Encoding Virulence Factors in Strains Isolated from the Pharmaceutical Production Environment

**DOI:** 10.3390/pathogens10020130

**Published:** 2021-01-27

**Authors:** Magdalena Ratajczak, Dorota Kaminska, Jolanta Dlugaszewska, Marzena Gajecka

**Affiliations:** 1Chair and Department of Genetics and Pharmaceutical Microbiology, Poznan University of Medical Sciences, 60-781 Poznan, Poland; mratajczak@ump.edu.pl (M.R.); dorotakaminska@ump.edu.pl (D.K.); jdlugasz@ump.edu.pl (J.D.); 2Institute of Human Genetics, Polish Academy of Sciences, 60-479 Poznan, Poland

**Keywords:** antimicrobial resistance, *Pseudomonas aeruginosa*, biofilm, pharmaceutical production environment

## Abstract

The spread of bacterial resistance to antibiotics affects various areas of life. The aim of this study was to assess the occurrence of *Pseudomonas aeruginosa*, and other bacteria mainly from orders *Enterobacterales* and *Staphylococcus* in the pharmaceutical production sites, and to characterize isolated strains in the aspects of antibiotic resistance, biofilm formation, and presence of genes encoding virulence factors. Genes encoding selected virulence factors were detected using PCR techniques. Antimicrobial susceptibility testing was applied in accordance with the EUCAST recommendations. A total of 46 *P. aeruginosa* strains were isolated and 85% strains showed a strong biofilm-forming ability. The qualitative identification of genes taking part in *Quorum Sensing* system demonstrated that over 89% of strains contained *lasR* and *rhlI* genes. An antimicrobial susceptibility testing revealed nine strains resistant to at least one antibiotic, and two isolates were the metallo-β-lactamase producers. Moreover, the majority of *P. aeruginosa* strains contained genes encoding various virulence factors. Presence of even low level of pathogenic microorganisms or higher level of opportunistic pathogens and their toxic metabolites might result in the production inefficiency. Therefore, the prevention of microbial contamination, effectiveness of sanitary and hygienic applied protocols, and constant microbiological monitoring of the environment are of great importance.

## 1. Introduction

Maintaining microbiological purity of the process of manufacturing medicinal products constitute an exceptionally important aspect in pharmaceutical production. The occurrence of bioburden on the surface of production equipment is a serious issue, because of the possibility of microbial contamination of the medicinal products. Any element of the pharmaceutical production environment: raw materials, technological media, personnel, air, or usable surfaces may constitute the source of microorganisms.

The dominant form of microbial existence, both in natural and industrial ecosystems, is biofilm in which microorganisms carry the ability to cause infections. In addition, bacteria in biofilms are more resistant to disinfectants and antibiotics in comparison to their planktonic counterpart [1,2,3]. In order to obtain a medicinal product of adequate microbiological quality, factories manufacturing medications follow the principles of Good Manufacturing Practice (GMP). Maintaining appropriate environmental conditions largely prevent the entry of microorganisms into medicinal products. Otherwise, they could be the cause of iatrogenic infections and cause deterioration of patients’ condition. In addition, the presence of microbes and their metabolites (toxins, enzymes) in medications may cause the loss of medicinal properties of medications due to the degradation of the active or preservative substances. Detection of the presence of pathogenic bacteria or exceeding the permissible quantitative limits in a medicinal product results in the non-release of the defective batch and the production suspension. Because of no legal regulations regarding the microbiological purity of dietary supplements, there is an actual risk of admission of contaminated preparations for sale, posing a threat to patients [4].

Elements of the water system and production equipment are the critical sites favouring colonization by bacteria and biofilm formation in the pharmaceutical production environment. Planktonic cells that break away from the structure of biological membranes can cause contamination of the produced medications, which results in failure to meet the required microbiological quality.

*P. aeruginosa* is an important human pathogen producing various virulence factors (e.g., exotoxin A, elastaze, protease IV). After entering the human body, bacteria adhere to host cells through adhesive type 4 fimbriae made of pilin protein. Endotoxin-lipopolysaccharide (LPS) constitutes an important factor in the pathogenicity of *P. aeruginosa*. The most important toxin secreted by *P. aeruginosa* is exotoxin A, a cytotoxin that has a lethal effect on eukaryotic cells, disrupting protein synthesis. It also affects mitochondria, causing an increase in membrane permeability and damages the electron transport system, thus causing breathing disorders. It plays an important role in the immune response of the host and the development of infection [5].

Proteolytic enzymes secreted by microorganisms play a significant role in the course of sepsis and eye infection due to the degradation of immunoglobulins and fibrin [6]. The presence of numerous virulence factors and formation of biofilm by *P. aeruginosa* is regulated through the mechanism of “bacteria to bacteria” communication—*Quorum Sensing* (QS) [2,6,7,8,9].

*P. aeruginosa* rods are characterized by multiple natural and acquired resistance to antibiotics [5,10,11,12]. An important mechanism of *P. aeruginosa* resistance is the production of class B metallo-β-lactamase (MBL). MBL positive strains are resistant or have a reduced sensitivity to penicillins, combinations of penicillins with β-lactamase inhibitors (no inhibitor effect), cephalosporins and carbapenems [10,11,12].

Additionally, other microorganisms isolated from the pharmaceutical production environment may have the ability to form biofilms and resistance to antibiotics. This resistance is widespread among clinical and environmental strains [13,14,15].

The aim of this study was to assess the occurrence of *P. aeruginosa* and other microbiota in the pharmaceutical production environment to characterize isolated strains in the aspects of antibiotic resistance, biofilm formation, and presence of genes encoding virulence factors.

## 2. Results

### 2.1. P. aeruginosa Dominated among the Isolated Bacterial Strains

In total, 81 bacterial strains were isolated in the A, B, and C factories, including 60 glucose non-fermenting bacterial rods, 14 strains from the order *Enterobacterales* and seven *Staphylococcus* spp. strains. Figure 1 shows growth of bacteria collected from the pharmaceutical production environments in factory A. *P. aeruginosa* were the most frequently observed in all factories (*n* = 46). Majority of the strains were isolated from wet areas. Thirty eightwere isolated from sinks, 30 from drains, and 13 from both work and device production surfaces. From the air in production rooms commonly observed microorganisms (such as *Bacillus* spp. and *Micrococcus* spp.) were isolated. They were not included in further analyses.

### 2.2. Majority of P. aeruginosa Strains Identified as Strong Biofilm Producers

On the basis of the value of absorbance, in accordance with the biofilm formation criteria all bacteria strains were tested towards their ability to form biofilm and then qualified into appropriate groups in terms of the biofilm forming rate. In total, 68% of all isolates have shown a strong biofilm-forming ability. A total of six strains, including *S. saprophyticus*, *M. luteus, C. freundii* and *P. stutzeri*, were found not to produce any biofilm. The majority of *P. aeruginosa* strains (*n* = 39) were found to be strong-biofilm producers (Appendix A). Figure 2 shows microscopic images of the biofilm formed on the PCV and stainless-steel surface. Biofilm formation was clearly observed for all tested strains and surfaces.

### 2.3. Genes Involved in the Bacteria Communication and the Virulence Factors Production in P. aeruginosa

The qualitative identification of genes taking part in QS system demonstrated that 89.1% of strains contained *lasR* gene, while *rhlI* and *rhlR* were present in 80.4% of *P. aeruginosa* isolates (Figure 3, Appendix A).

The largest number of strains contained a *toxA,* gene responsible for production of exotoxin A (97.8%). *AprE* gene responsible for production of elastase was present in 43 out of 46 (93.4%) tested strains. The least isolates contained a gene responsible for production of protease IV (39.1%) (Figure 3, Appendix A).

### 2.4. Correlation of the Ability to Create Biofilms and the Presence of Genes Encoding Virulence Factors and QS

Appendix A contain a summation of results of the qualitative identification of genes taking part in the QS and encoding virulence factors, and a biofilm formation rate in *P. aeruginosa* strains isolated from the evaluated pharmaceutical environment. A total of nine isolates contained all of the tested genes and were characterized by the strong ability to form biofilm. In total, two strains did not contain any of the identified QS genes, one of tested strains was a moderate producer of biofilm and the other was characterized by poor ability to form biomass. The majority of strains (67.4%) contained all of the tested genes related to QS system and simultaneously were strong biofilm producers. Eleven isolates contained all of the identified genes encoding virulence factors and simultaneously were strong biofilm producers.

### 2.5. The Pharmaceutical Production Environment Might Be a Source of Antibiotics Resistant Bacteria

A total of nine *P. aeruginosa* strains out of 46 analysed strains demonstrated resistance to at least one antibiotic. In total, eight strains were resistant to one of the tested aminoglycosides. The largest percentage of sensitive strains was detected towards ciprofloxacin (95.6%) and ticarcillin with clavulanic acid (97.8%). Among beta-lactam antibiotics the largest percentage of resistant strains was detected towards ceftazidime (6.5%) and imipenem (4.3%). Among the order *Enterobacterales*, 35.7% were tobramycin resistant. In total, 21.4% of strains were resistant to ceftazidime and gentamycin. All strains were sensitive to ciprofloxacin and piperacillin. Figure 4 shows the antibiotic susceptibility test results using the disc diffusion method and detection of MBL with the disc diffusion synergy test. Data of antimicrobial resistance of the tested strains is presented in Table 1.

Tests detecting carbapenemases formation were carried out for strains that turned out to be resistant or intermediate to imipenem and/or meropenem. Two isolates of the following turned out to be producers of metallo-beta-lactamases MBL; no strain was a producer of carbapenemases KPC. In the aspect of places of *P. aeruginosa* strains isolation, it was found that strains resistant to the tested antibiotics were isolated from the aquatic environments. The largest number of strains resistant to at least one antibiotic were isolated from sinks. Additionally, strains that turned out to be MBL producers were isolated from the aquatic environments. One strain was a producer of the extended spectrum β-lactamase.

Out of 46 *P. aeruginosa* strains, six isolates (13.0%) resistant to gentamicin, three isolates (6.5%) resistant to ceftazidime, three isolates (6.5%) resistant to tobramycin, two isolates (4.3%) resistant to imipenem and one (2.2%) resistant to ticarcilin/clavulanate, meropenem and ciprofloxacin were observed. Among strains from the order *Enterobacterales* five (35.7%) were resistant to tobramycin, three (21.4%) were resistant to ceftazidime and gentamicin, and one (7.1%) to ticarcilin/clavulanate. There were no observed resistance to cefoxitin, erythromycin and clindamycin among the *Staphylococcus* spp. strains.

## 3. Discussion

Medicines and dietary supplements may be contaminated by wide spectrum of microbes. Medicines must meet the microbiological purity criteria in accordance with applicable legal regulations [16].

The European Medicines Agency (EMA) is the European institution that supervises the quality of medicines and ensures public health safety in the European Union. The agency, among others, is responsible for the coordinating of withdrawals actions of the drug, which that do not meet the requirements, across the EU. In 2018, the agency received 147 suspected quality defect notifications. Of these, 123 cases were confirmed quality defects and led to batch recalls of 27 centrally authorized medicines. Among these products, 5 were withdrawn from the market due to microbial contamination [17]. A much larger number of drugs that do not meet the requirements for microbiological purity are withdrawn in the USA than in Europe. FDA withdrew 74 products from the market due to microbiological contamination in 2018, and from 1 January 2019 to 21 June 2019 as many as 459 products. The main reasons for the recall of the drugs were being out of specification for mould, yeast and bacteria, and were being found to contain microbial contaminants *S. aureus*, *S. saprofhyticus* and *Burgholderia cepatia*, *Burgholderia multivorans*, *P. aeruginosa*, *P. brenneri*, *P. fluroescens, Paecilomyces saturates* and *Aspergillus fumigatus* [18].

Jimenez L. found that FDA product recall data for 134 non-sterile pharmaceutical products demonstrated that 48% of recalls were due to contamination by either *B. cepacia*, *Pseudomonas* spp., or *Ralstonia picketti*, while yeast and mould contamination were found in 23% of recalls. Gram-negative bacteria accounted for 60% of recalls, but only 4% were associated with Gram-positive bacteria [19].

In our previous research, the microbiological contamination that exceeded the limit was found in 1.3% of medicines and 14.3% dietary supplements. The products were mainly contaminated by *Enterobacterales*. None of the samples was found to contain bacteria with a higher disease-causing potential such as, for example, *Salmonella* spp. The presence of bacteria from the order *Enterobacterales*, chiefly *E. coli*, points to contamination of fecal origin and may indicate that the hygienic conditions in the manufacturing plant were inadequate. The performed microbiological control of the completed pharmaceutical products have prevented the release of contaminated product series for sale [16].

A persistent level of product recall for microbial contamination highlights the risk for quality of drugs and for health and life of patients. Contaminated with microbes, pharmaceutical products may be the cause of drug-induced infections. In 2004, a large outbreak of catheter-associated Gram-negative BSIs among patients in an oncology chemotherapy centre was recorded with the 27 confirmed positive blood cultures cases: 20 for *E. cloacae*, 2 for *K. oxytoca*, 5 for both. The reason for this outbreak was the injection of the isotonic sodium chloride solution contaminated with *K. oxytoca, E. cloacae,* through the central venous catheters of patients [20,21].

Raw materials, pharmaceutical production environment and personnel are the sources of microbial contaminations. Previously, analysing biological contamination of humid pharmaceutical production environments, such as sink traps or sewage grates, it was found that in 26% of tested samples Gram-negative rods of *Pseudomonas*, *Stenotrophomonas*, *Acinetobacter*, *Aeromonas*, and rods from the order *Enterobacterales* were identified. *Pseudomonas* spp. were the most frequently isolated and the dominant species were *Pseudomonas aeruginosa* [22].

In 2018, the North Carolina–based King Bio Inc. recalled from the market all of its water-based products for human and animal use due to high levels of microbial contamination. Several microbial contaminants were found in products, including *P. brenneri*, *P. fluorescens* and *B. multivorans*—microbes that can cause illness in people with compromised immune systems. What is more, evidence collected during the FDA’s inspection indicated recurring microbial contamination associated with the water system used to manufacture drug products [23].

Tršan et al., reported a catalogue of cleanroom microorganisms isolated from four different cleanrooms which indicated that 78% of isolated bacteria were Gram-positive and majority of the species (>70%) were belonging to the normal human microbiota. The microorganisms were derived from the air (10–15%, *Bacillus* spp., fungi) and water (5–10%, Gram-negative microorganisms). Additionally, in our study, most strains were isolated from wet areas. Non-fermenting rods and bacteria from the order *Enterobacterales* were dominant. The source of Gram-negative microorganisms in the pharmaceutical production environment are usually water systems [24].

The fact that bacteria form biological membranes poses a serious problem for the health care system, and both food production and pharmaceutical industry [10,25,26,27]. We demonstrated that 100% of *P. aeruginosa* strains isolated from the pharmaceutical production environment were able to form biofilm with the greatest number of stains displaying high degree of biofilm formation. In accordance with the previously published data, Marchand et al., provided evidence that the *P. aeruginosa* strain, which formed biofilm in a food environment, communicated with other bacteria e.g., with *Listeria monocytogenes,* forming a multi-species biofilm [28]. Additionally, it is believed that 80–100% of the strains isolated from clinical infections are capable of forming biofilm [29,30]. In our study, after analysing the location of those bacteria that displayed high degree of biofilm formation, it has been observed that most of the isolates originated from sinks and drains. Humid sites in the pharmaceutical production environment are critical sites that promote the formation of biofilms.

Bacteria cells from the *Pseudomonas* genus have two QS systems. The *las* system which involves *lasR* and *lasI* gene and the *rhl* system which involves of *rhlR* and *rhlI* genes [31]. Here we observed that only three isolates of *P. aeruginosa* did not contain any of the analysed genes. On the other hand, most of them contained all of the assessed QS genes. Similar results have been reported on the strains of *P. aeruginosa,* isolated from respiratory tract infections, showing that 68.7% of isolates contained all four genes; 18.7% of the strains, on the other hand, did not contain any of the analysed genes [32]. In another report on strains from different clinical materials, 81.6% of the isolates contained at least one of the analysed genes [33].

While comparing the content of genes responsible for QS in tested *P. aeruginosa* strains and the capability of these isolates to form biofilms, we observed that almost all of them were characterized with high capability to form biofilm. On the other side, among the strains which did not contain any of the QS genes, two isolates formed biofilm to a moderate extent, and one displayed a poor capability of forming biological membrane.

Here, we have focused on several of virulence factors produced by *P. aeruginosa* strains. Here, the presence of the gene responsible for production of elastase has been observed in majority of examined strains. Alkaline protease enables growth of bacteria in the organism of an infected host and influence its immune system. In our study, in more than half of the analysed environmental strains, a gene determining production of alkaline protease was found. The *lysyl*, gene encoding peptidase IV was also present in the tested *P. aeruginosa* strains. Our results of the identification of the *toxA* gene, responsible for production of exotoxin A, seem to resemble the previously published data [31,34].

According to our research results, *P. aeruginosa* strains isolated from the pharmaceutical production environment have contained genes responsible for the production of virulence factors in very similar proportions comparing to strains isolated from infections [30,31].

Our evaluation concerning the drug susceptibility of the *P. aeruginosa* and *Enterobacterales* strains isolated from the environment of pharmaceutical production indicated that 6.5% of the *P. aeruginosa* strains were resistant to ceftazidime. Among the tested *P. aeruginosa* strains, there was one resistant and one with medium resistance to ciprofloxacin. More strains displayed resistance to gentamicin (13.0%) and tobramycin (6.5%). Moreover, it was demonstrated that environmental strains may display resistance to carbapenems. Among the analysed strains, one displayed resistance and one intermediate to meropenem. Similarly, in the in case of imipenem, two strains were resistant and one intermediate. One strain was resistant to both carbapenems and other antibiotics tested. Frequency of Gram-negative bacteria’s resistance to carbapenems has been increasing all over the world. Acquisition of MBL-coding genes is considered one of the most significant carbapenems resistance mechanisms [10,35,36,37]. We found that two strains (4.3%) were MBL positive. One strain that was isolated from the production facility sink, was resistant to all the tested antibiotics; additionally, this bacterium was characterized by additional MBL resistance mechanism. Another strain, in which metallo-β-lactamase formation was observed, was isolated from a floor drain.

*Enterobacterales* strains were also found here to be resistant to antibiotics. Among the tested strains, resistance to tobramycin, ceftazidime and gentamycin were observed. All strains were sensitive to ciprofloxacin and piperacillin. Moreover, one strain produced extended spectrum β-lactamase.

We observed that strains isolated from the pharmaceutical production environment were far less resistant to antibiotics than the strains isolated from clinical materials. According to the research carried out in 36 countries (Latin America, Asia, Africa and Europe) by the International Nosocomial Infection Control Consortium (INICC), 47.2% of *P. aeruginosa* strains isolated from infections were resistant to imipenem [20]. Tests done on strains isolated from patients hospitalized due to pneumonia in the Unites States and Europe showed that, respectively, 23.7 and 34.2% of strains were resistant to meropenem.

Summarizing, the bacterial strains isolated at the manufacturing premises have the ability to produce biofilm, which leads to problems with microbiota eradication. Moreover, the majority of *P. aeruginosa* strains isolated from the pharmaceutical production environment was characterized by the presence of genes encoding various virulence factors. This study identified *P. aeruginosa* strains exhibiting resistance to routinely used antibiotics in the pharmaceutical environment. Therefore, the results demonstrated that the pharmaceutical production environment is a source of potentially pathogenic microorganisms which might contaminate drugs manufactured on the premises. The prevention of microbial contamination, and adequate manufacturing and storage conditions are of great importance. The environment of pharmaceutical production and the effectiveness of sanitary and hygienic methods should be persistently monitored to reduce contamination of medicinal products and eliminate sources of potentially pathogenic microorganisms.

## 4. Material and Methods

Microbiological purity of the pharmaceutical production environment in three different drug factories, that have EU GMP accreditation, localized in Poland was assessed. These factories produce non-sterile and finished dosage forms of drugs. Samples were collected during batch production as part of the previously implemented sanitary and hygienic procedures. Smears were taken from the production devices, work surfaces, sinks, and drains in the production rooms. Bacterial strains were isolated from the taken smears and underwent further tests. The microbiological purity of the air was investigated using the impact method in accordance with GMP [38]. In addition, numerous medicinal products manufactured in those pharmaceutical production environments were assessed in accordance with the European Pharmacopoeia 9.0 [39].

### 4.1. Isolation of Bacteria Strains

Samples were collected from the tested surfaces (25 cm^2^), inside of a tap, drain traps and drain gratings using sterile swabs moistened in a sterile NaCl solution and/or using contact plates (Oxoid, Wesel, Germany). Identification to the level of genus was possible based on microscopic and macroscopic assessment of bacterial colonies as well as biochemical tests: ID Color Catalase (BioMérieux, Marcy-l’Étoile, France)—the reagent detects the presence of catalase, thus enabling the differentiation of bacteria that possess this characteristic, Bactident Oxidase test (MERCK, Darmstadt, Germany)—for the detection of cytochrome oxidase in microorganisms. Identification of the isolated strains to the level of species was carried out using an automatic identification system VITEK^®^ Compact 2 (BioMérieux, Durham, NC, U.S.). GN cards (BioMérieux, Durham, NC, U.S.) were used for identification of *Enterobacterales* and a select group of nonfermenting Gram-negative organisms, GP cards (BioMérieux, Durham, NC, U.S.) were applied for the identification of enterococci, streptococci, staphylococci and a selected group of gram-positive organisms. All identified strains were stored for further tests in temperature −80 ± 10 °C, in microbanks (Pro-Lab Diagnostics, Richmond Hill, ON, Canada).

### 4.2. Quantitative Assessment of Biofilm Formation in Vitro

Biofilm formation was determined by the microtiter plate assay, as previously reported [40]. Briefly, three wells of 96-well flat-bottomed plastic plate were filled with 200 µL of each tested bacterial suspension. The plates were covered and incubated aerobically for 24 h at 37 °C. Then, the content of each well was aspirated, and each well was washed three times with 250 µL of sterile physiological saline. Then, plates were stained for 15 min with 0.2 mL. of 2% crystal violet. Then, each well was washed with physiological saline solution. After the plates were air dried, 200 µL of 99% methanol per well were added, absorbance of each well was measured at 590 nm, using the Infinite M200 (Tecan, Grödig, Austria). Simultaneous biofilm formation assay on the PCV and stainless steel surface was carried. Biofilm was grown using PCV and stainless steel discs. The discs inoculated with bacterial suspension in BHI were incubated at 36 °C for 48 h. After incubation, the medium was removed, and the discs were gently washed twice with distillated water dehydrated and fixed in increasing concentrations of ethanol. Scanning electron microscopy (SEM) was used to illustrate the study results.

### 4.3. Identification of Genes Taking Part in QS and Encoding Virulence Factors in P. aeruginosa Strains

DNA extraction from the isolated *P. aeruginosa* strains was carried out using the in house developed thermal extraction method and the obtained DNA was stored for further analysis in temperature −20 ± 2 °C.

For the qualitative assessment of genes encoding virulence factors and taking part in QS eight PCR reaction starters designed using the Primer3 programme (available at http://bioinfo.ut.ee/primer3-0.4.0) and checked in terms of appropriate parameters using Primer-BLAST function (Basic Local Alignment Search Tool) available in an online base NCBI (National Center for Biotechnology Information) were used. The primers for the tested genes (*lasI*, *lasR*, *rhlI*, *rhlR*, *aprE*, *lasB*, *lysyl*, *toxA*) were designed based on the *P. aeruginosa* PAO1 genomic sequence PAO1nr Ac. Num. NC_002516.2. Appendix A presents the applied starters and PCR conditions applied (Appendix A). The presence or absence of amplified fragments of taking part in QS (*lasI*, *lasR*, *rhlI*, *rhlR*) and the virulence factors (*aprE*, *lasB*, *lysyl*, *toxA*) gene sequences were analysed.

### 4.4. Evaluation of Antibiotic Susceptibility of the Isolated Bacterial Strains

Assessment of antibiotics susceptibility of *P. aeruginosa* and other isolated bacterial strains was carried out using the disc diffusion method in accordance with the EUCAST recommendations [41]. Gentamicin 10 µg, ciprofloxacin 5 µg, ceftazidime 10 µg, tobramycin 10 µg, ticarcillin with clavulanic acid 75/10 µg, imipenem 10 µg, and meropenem 10 µg were used to test the drug susceptibility for *P. aeruginosa* (Figure 2). For the order *Enterobacterales*, gentamicin 10 µg, ciprofloxacin 5 µg, ceftazidime 10 µg, tobramycin 10 µg and piperacillin 30 µg were used. Erythromycin 15 µg, clindamycin 2 µg and cefoxitin 30 µg were examined in staphylococci.

### 4.5. Detection of Mechanisms of Resistance to Antibiotics

Carba NP test and phenotypic screening test with EDTA for metallo-beta-lactamases (MBL) as well as a test with boronic acid for carbapenemases (KPC) were carried out to detect the medium sensitive non-fermenting strains or resistant to carbapenemases and resistant to ticarcillin with clavulanic acid non-fermenting strains.

For all isolates of *Enterobacterales,* an extended-spectrum beta-lactamases (ESBL) using a double-disc synergy test were assessed. *Staphylococcus* spp. resistance to methicillin was determined using the disc with cefoxitin 30 µg.

## Figures and Tables

**Figure 1 pathogens-10-00130-f001:**
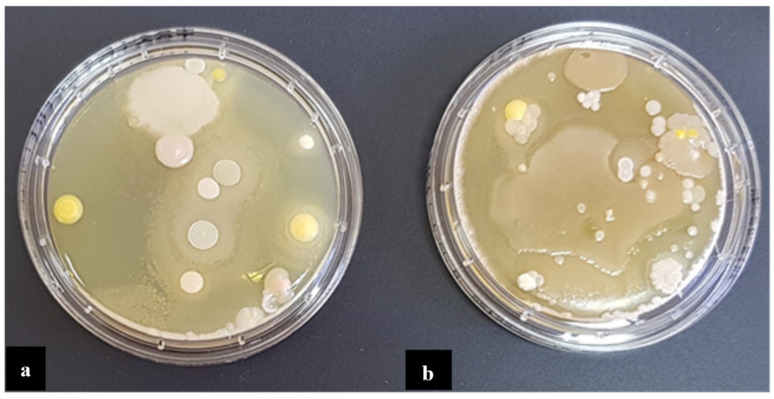
Bacteria identified on the production equipment (**a**), and in the air (**b**), in the pharmaceutical production site A.

**Figure 2 pathogens-10-00130-f002:**
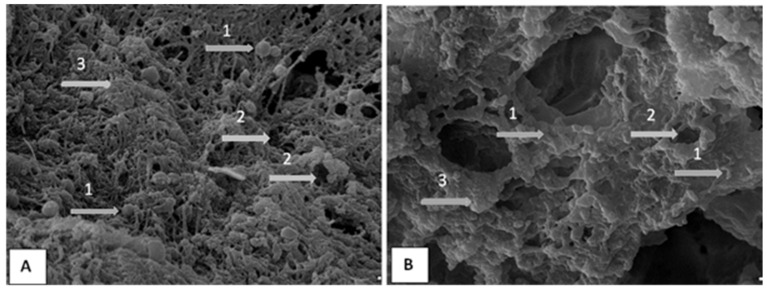
Scanning electron microscopy (SEM) image of biofilm on the PCV (**A**) and stainless steel (**B**) surface. Bacteria (1) and water channels (2) are clearly visible and some ‘towers’ (3) as well.

**Figure 3 pathogens-10-00130-f003:**
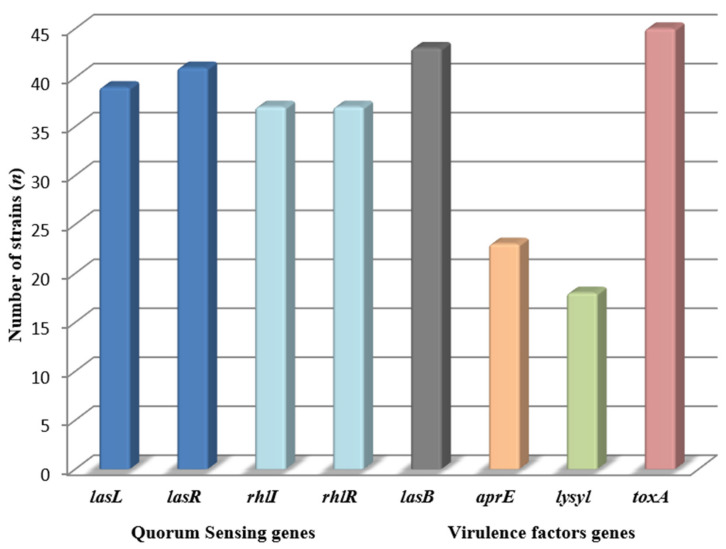
Presence of QS and virulence factors genes in *P. aeruginosa* (*n* = 46) strains.

**Figure 4 pathogens-10-00130-f004:**
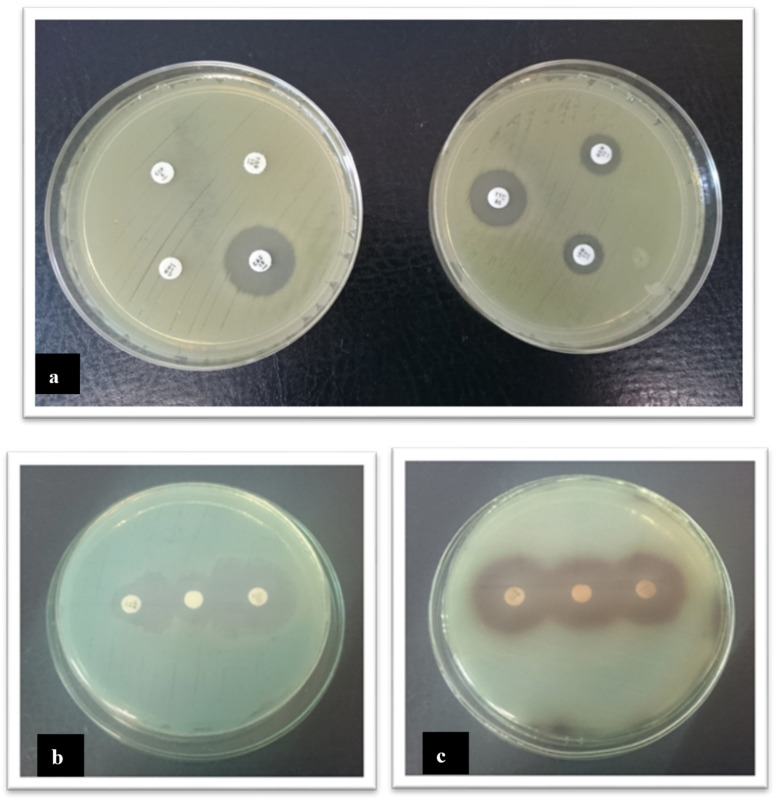
Evaluation of antibiotic susceptibility of *P. aeruginosa* using the disc diffusion method (**a**) and phenotypic detection of MBL with the disc diffusion synergy test: positive (**b**), and negative (**c**). Enhancement of the zone of inhibition in the area between imipenem/EDTA and ceftazidime/EDTA disks was interpreted as a positive result for MBL production.

**Table 1 pathogens-10-00130-t001:** Antimicrobial resistance among tested strains.

		Cefoxitin	Ceftazidime	Piperacylina	Ticarcilin/ Clavulanate	Imipenem	Meropenem	Gentamicin	Tobramycin	Clindamycin	Erythromycin	Ciprofloxacin
*P. aeruginosa* (*n* = 46)	R (%)	-	6.5	-	2.2	4.3	2.2	13.0	6.5	-	-	2.2
	S (%)	-	93.5	-	97.8	93.5	95.6	87.0	93.5	-	-	95.6
*Enterobacterales* (*n* = 14)	R (%)	-	21.4	0.0	7.1	-	-	21.4	35.7	-	-	0.0
	S (%)	-	78.6	100	92.9	-	-	78.6	64.3	-	-	100
*Staphylococcus* spp. (*n* = 7)	R (%)	0.0	-	-	-	-	-	-	-	0.0	0.0	-
	S (%)	100	-	-	-	-	-	-	-	100	100	-

R: resistant; S: susceptible.

## Data Availability

Not applicable.

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
