# Peer review of "Antibiotic Resistance, Biofilm Formation, and Presence of Genes Encoding Virulence Factors in Strains Isolated from the Pharmaceutical Production Environment"

_pathogens, 2021, doi:10.3390/pathogens10020130_

Round 1

Reviewer 1 Report

In this manuscript Ratajczak et al. demonstrate the presence of many clinically significant bacterial genus in pharmaceutical production sites. Specifically, they show that pharmaceutical sites are predominantly contaminated by strains of Pseudomonas aeruginosa (Pa), which are strong biofilm producers and exhibit drug-resistance profiles. Chiefly, these Pa strains contain many virulence factors which can contaminate the drug production pipeline and can pose serious threats to public health. Overall, this study is well presented and important. My only suggestion is that the information in supplementary table S1 could be presented in a bar graph or Venn diagram format in the main text. This could possibly be done in a way to convey the number of strains that include the specific QS and virulence genes, which I think would make the manuscript more impactful.

Author Response

Reviewer 1

In this manuscript Ratajczak et al. demonstrate the presence of many clinically significant bacterial genus in pharmaceutical production sites. Specifically, they show that pharmaceutical sites are predominantly contaminated by strains of Pseudomonas aeruginosa (Pa), which are strong biofilm producers and exhibit drug-resistance profiles. Chiefly, these Pa strains contain many virulence factors which can contaminate the drug production pipeline and can pose serious threats to public health. Overall, this study is well presented and important.

My only suggestion is that the information in supplementary table S1 could be presented in a bar graph or Venn diagram format in the main text. This could possibly be done in a way to convey the number of strains that include the specific QS and virulence genes, which I think would make the manuscript more impactful.

Dear Reviewer,

Thank you for your comments and suggestion to improve the manuscript content. We appreciate your participation in reviewing our paper.

We have taken into consideration the suggestion  and we've added Figure 3 (Presence of QS and virulence factors genes in P. aeruginosa strains) in the main text (page 4, Figure 3).

Reviewer 2 Report

The manuscript entitled “Antibiotics resistance, biofilm formation, and presence of genes encoding virulence factors in strains isolated from he pharmaceutical production environment” have demonstrated the occurrence of P. aeruginosa and other microbiota in the pharmaceutical production environment to characterize isolated strains in the aspect of antibiotic resistance, biofilm formation and presence of genes encoding virulence factor. The subject of the manuscript is important. Research increasingly shows that bacteria are increasingly resistant to antibiotics

Authors should correct manuscript according to the suggestions.

Minor issues:

Lin 194 – 196: dots are missing in names of bacteria

References:

Authors should checked and corrected Reference no 3 according to journal guidelines

Author Response

Reviewer 2

The manuscript entitled “Antibiotics resistance, biofilm formation, and presence of genes encoding virulence factors in strains isolated from he pharmaceutical production environment” have demonstrated the occurrence of P. aeruginosa and other microbiota in the pharmaceutical production environment to characterize isolated strains in the aspect of antibiotic resistance, biofilm formation and presence of genes encoding virulence factor. The subject of the manuscript is important. Research increasingly shows that bacteria are increasingly resistant to antibiotics

Authors should correct manuscript according to the suggestions.

Dear Reviewer,

Thank you for your comments. We changed the manuscript text as suggested.

Minor issues:

Lin 194 – 196: dots are missing in names of bacteria

We corrected (line 205-207, page 9)

References:

Authors should checked and corrected Reference no 3 according to journal guidelines

We corrected this reference (line 417-420 , page 13)

Reviewer 3 Report

Antibiotic resistance is a growing global menace. In this manuscript, Ratajczak et al. describe bacterial strains isolated from the pharmaceutical production environment in Poland. This work will be of broad interest to readers of the pathogens journal. However, some items need to be addressed before publication.

  1. My primary concern is the sample collection. The authors have collected samples from production devices (a), work surfaces (b), sinks (c), drains (d), and air (e) (Line 303-305). Figure 1 shows the bacteria identified from production devices (a) and the air (e). However, samples from a and e are not included in 81 bacterial strains (38 from c, 30 from d, and 13 from b). The authors should clarify it.
  2. Line 106: The authors claim that over 89% of strains contained lasR and rhlI gene. However, Table S1B shows rhlI only present in 80.4% P. aeruginosa strain. Besides, the authors should replace “Supplementary Table S2 A and B” (line 107 and 111) with “Supplementary Table S1 B”.
  3. Line 114: The author should replace “supplementary Table S2” with “supplementary Table S1”.
  4. Since the authors have detected quorum sensing genes and virulence factors genes by PCR. It would be nice to detect which MBLs are presented by PCR as well.

Author Response

Reviewer 3

Antibiotic resistance is a growing global menace. In this manuscript, Ratajczak et al. describe bacterial strains isolated from the pharmaceutical production environment in Poland. This work will be of broad interest to readers of the pathogens journal. However, some items need to be addressed before publication.

Dear Reviewer,

Thank you for your constructive comments and specific suggestions to improve the manuscript content. We appreciate your participation in reviewing our paper.

  1. My primary concern is the sample collection. The authors have collected samples from production devices (a), work surfaces (b), sinks (c), drains (d), and air (e) (Line 303-305). Figure 1 shows the bacteria identified from production devices (a) and the air (e). However, samples from a and e are not included in 81 bacterial strains (38 from c, 30 from d, and 13 from b). The authors should clarify it.

We changed the sentence “Most strains were isolated from wet areas: 38 were isolated from sinks, 30 from drains, and 13 from the work surface.” to “Majority of the strains were isolated from wet areas. Thirty eight were isolated from sinks, 30 from drains, and 13 from both work and device production surfaces.” (line 88-90, page 2)

We have added an explanatory sentence: “From the air in production rooms commonly observed microorganisms (such as Bacillus spp. and Micrococcus spp.) were isolated. They were not included in further analyzes.” (line 90-92, page 2)

  1. Line 106: The authors claim that over 89% of strains contained lasR and rhlI gene. However, Table S1B shows rhlI only present in 80.4% P. aeruginosa strain. Besides, the authors should replace “Supplementary Table S2 A and B” (line 107 and 111) with “Supplementary Table S1 B”.

We corrected this sentence to “The qualitative identification of genes taking part in QS system demonstrated that 89.1% of strains contained lasR gene, while rhlI and rhlR were  present in 80.4% of P. aeruginosa isolates.”(line110-112, page4)

We changed ,,Supplementary Table S2 A and B” to ,,Supplementary Table S1 B” (line 112 and 116, page 4).

  1. Line 114: The author should replace “supplementary Table S2” with “supplementary Table S1”.

We changed “Supplementary Table S2” to “Supplementary Table S1” (line 124, page 4).

  1. Since the authors have detected quorum sensing genes and virulence factors genes by PCR. It would be nice to detect which MBLs are presented by PCR as well.

Thank you for this comment. We detected metallo-beta-lactamases using  two different phenotypic methods which  are currently being used in clinical laboratories. We will take this suggestion into account in our further research.